# Intranasal Delivery of RGD-Containing Osteopontin Heptamer Peptide Confers Neuroprotection in the Ischemic Brain and Augments Microglia M2 Polarization

**DOI:** 10.3390/ijms22189999

**Published:** 2021-09-16

**Authors:** Dashdulam Davaanyam, Il-Doo Kim, Ja-Kyeong Lee

**Affiliations:** Department of Anatomy, Inha University School of Medicine, Incheon 22212, Korea; 22192309@inha.edu (D.D.); ilk4001@med.cornell.edu (I.-D.K.)

**Keywords:** osteopontin, stroke, M2 microglia, heptamer, RGD

## Abstract

Osteopontin (OPN), a phosphorylated glycoprotein, is induced in response to tissue damage and inflammation in various organs, including the brain. In our previous studies, we reported the robust neuroprotective effects of the icosamer OPN peptide OPNpt20, containing arginine-glycine-aspartic acid (RGD) and serine-leucine-alanine-tyrosine (SLAY) motifs, in an animal model of transient focal ischemia and demonstrated that its anti-inflammatory, pro-angiogenic, and phagocytosis inducing functions are responsible for the neuroprotective effects. In the present study, we truncated OPNpt20 to 13 or 7 amino acid peptides containing RGD (R) and/or SLAY (S) motifs (OPNpt13RS, OPNpt7R, OPNpt7RS, and OPNpt7S), and their neuroprotective efficacy was examined in a rat middle cerebral artery occlusion (MCAO) model. Intranasal administration of all four peptides significantly reduced infarct volume; OPNpt7R (VPNGRGD), the 7-amino-acid peptide containing an RGD motif, was determined to be the most potent, with efficacy comparable to that of OPNpt20. Additionally, sensory–motor functional deficits of OPNpt7R-administered MCAO animals were significantly improved, as indicated by the modified neurological severity scores and rotarod test. Notably, the expression of M1 markers was suppressed, whereas that of M2 markers (Arginase 1, CD206, and VEGF) was significantly enhanced in OPNpt7R-treated primary microglia cultures. Inflammation resolution by OPNpt7R was further confirmed in MCAO animals, in which upregulation of anti-inflammatory cytokines (Arg1, IL-10, IL-4, and CD36) and enhanced efferocytosis were detected. Moreover, studies using three mutant peptides (OPNpt7R-RAA or OPNpt7R-RAD, where RGD was replaced with RAA or RAD, respectively, and OPNpt7R-sc containing scrambled sequences) revealed that the RGD motif plays a vital role in conferring neuroprotection. In conclusion, the RGD-containing OPN heptamer OPNpt7R exhibits neuroprotective effects in the post-ischemic brain by suppressing M1 markers and augmenting M2 polarization of microglia and the RGD motif plays a critical role in these activities.

## 1. Introduction

Osteopontin (OPN) is a multifunctional glycoprotein that is highly phosphorylated and expressed in numerous cell types. OPN plays important roles in various cell types by interacting with multiple receptors, including integrins and CD44 variants [1]. Protective effects of OPN have been reported in several central nervous system diseases, including Parkinson’s disease [2,3], multiple sclerosis [4,5], and subarachnoid or intracerebral hemorrhage [6,7]. Since Meller (2005) [8] first reported the protective effects of OPN in middle cerebral artery occlusion (MCAO) using OPN-deficient mice, numerous reports have demonstrated the neuroprotective effects of OPN in various stroke animal models [9,10,11]. In a previous study, we demonstrated a robust neuroprotective effect of recombinant OPN in a rat MCAO model and enhanced efficiency by encapsulating the recombinant OPN in biodegradable gelatin microspheres [11].

OPN possesses two specific integrin-binding motifs that are known as arginine-glycine-aspartic acid (RGD) and serine-leucine-alanine-tyrosine-glycine-leucine-arginine (SLAYGLR) motifs. Administration of OPN peptides that are 20 or even 10 amino acids in length and contain RGD and SLAYGLR motifs can significantly reduce infarct volume in MCAO animal models [12], and a similar 15 amino acid peptide also exerts neuroprotective effects in the rat substantia nigra following toxic insult [2]. In our previous studies, we reported that an OPN icosamer peptide (OPNpt20) containing RGD and SLAYGLR motifs exerted anti-inflammatory effects in the post-ischemic brain [13,14]. Recently, we also demonstrated that the same peptide confers pro-angiogenic effects in the post-ischemic brain [15].

In the post-ischemic brain, microglia are rapidly activated to differentiate into either the M1 or M2 phenotype, where proinflammatory functions are attributed to M1, and more anti-inflammatory and protective functions are attributed to M2 phenotypes ([16,17]; for review, see [17]). OPN augments the M2 microglia response in cerebral ischemia [18], and this phenotype represents a relatively benign activation state that accelerates inflammation resolution by removing dead cell debris and toxic substances and facilitates the repair of damaged tissue and recovery following stroke. In particular, a critical role of OPN in the phagocytic function of macrophages has been reported in various disease models. For example, OPN secreted by brain macrophages has been implicated in the phagocytosis of fragmented dead cell debris in a rat MCAO model [19] and in an Alzheimer’s disease mouse model [20]. Recently, we demonstrated that an RGD-containing 7-amino-acid OPN peptide (OPNpt7R, VPNGRGD) enhances the motility and phagocytic activity of microglia via the Fak, Erk, and Akt signaling pathways [21]. The aim of our current study is to investigate if OPNpt7R plays a role in M2 polarization of microglia and confers neuroprotection in the ischemic brain.

## 2. Results

### 2.1. OPN Heptamer Peptides (OPNpt7) Containing an RGD Motif Confers a Neuroprotective Effect in the Post-Ischemic Rat Brain

To compare the neuroprotective effects of various OPN peptides containing the RGD motif (OPNpt20, OPNpt13R, or OPNpt7R), containing the SLAY motif (OPNpt7S), or containing both the RGD and the SLAY motifs (OPNpt7RS) in the post-ischemic brain, each peptide (Figure 1A, 500 ng/rat) was administered intranasally at 1 h post-MCAO, and infarct volumes were measured at 48 h post-MCAO when the infarct volume reacheed the maximum level. Mean infarct volumes were reduced to 45.5 ± 5.2% (*n* = 11, *p* < 0.01), 55.3 ± 8.5% (*n* = 5, *p* < 0.01), 46.4 ± 5.2% (*n* = 13, *p* < 0.01), 66.3 ± 9.7% (*n* = 6, *p* < 0.01), and 73.8 ± 8.5% (*n* = 4, *p* < 0.05), respectively, in OPNpt20-, OPNpt13R-, OPNpt7R-, OPNpt7RS-, or OPNpt7S-administered MCAO groups compared to the values for PBS-treated MCAO controls (Figure 1B,C). These results indicate that the neuroprotective effect of OPNpt7R is comparable to that of OPNpt20 or OPNpt13; however, this effect is significantly greater than that of OPNpt7RS or OPNpt7S.

### 2.2. RGD Motif Plays a Critical Role in OPNpt7R-Mediated Infarct Suppression in the Post-Ischemic Brain

Next, we examined the importance of the RGD motif in OPNpt7R. Three mutant peptides, OPNpt7R-RAA (RGD was replaced with RAA), OPNpt7R-RAD (RGD was replaced with RAD), or OPNpt7R-sc (scrambled OPNpt7R sequence) (Figure 1A), were administered 1 h post-MCAO, and infarct volume was examined at 48 h post-MCAO. Mean infarct volumes were 78.8 ± 6.9% (*n* = 4), 102.8 ± 8.1% (*n* = 4), and 78.6 ± 6.3% (*n* = 4), respectively, in the OPNpt7R-RAD-, OPNpt7R-RAA-, or OPNpt7R-sc-administered MCAO groups, and they were significantly higher than those in the OPNpt7R-administered MCAO group (Figure 1D,E). These results indicate a critical role for the RGD motif in OPNpt7R-mediated infarct suppression in the post-ischemic brain.

### 2.3. OPNpt7 Exhibits a Wide Therapeutic Time Window in the Post-Ischemic Brain

To examine the neuroprotective efficacy of OPNpt7R in more detail, 100 ng, 500 ng, or 1 mg doses of OPNpt7R were administered intranasally at 1 h post-MCAO. The mean infarct volumes assessed 2 days after MCAO were reduced to 82.5 ± 6.2% (*n* = 3, *p* < 0.01), 46.4 ± 5.2% (*n* = 13, *p* < 0.01), and 38.9 ± 8.2% (*n* = 3, *p* < 0.01), respectively, compared to those of PBS-treated MCAO controls, indicating that OPNpt7R dose-dependently suppressed infarct volume (Figure 2A,B). Additionally, when 500 ng of OPNpt7R was administered at 6 or 1 h prior to- or 1 or 6 h post-MCAO, the mean infarct volumes were significantly suppressed in all four groups compared to those of the PBS-treated MCAO controls, where the greatest effect was observed at 6 h pre-treatment (Figure 2C,D). Importantly, 6 h post-administration was also able to induce a significant neuroprotective effect (Figure 2C,D). These results indicate that OPNpt7R possesses a wide therapeutic time window.

### 2.4. OPNpt7 Ameliorates Neurological Deficits and Motor Impairments

The mean modified neurological severity score (mNSS) of treatment-naïve MCAO control group was 11.6 ± 0.3 (*n* = 10) at 2 days post-MCAO. The mNSSs of the MCAO+OPNpt7R (500 ng) group (7.1 ± 0.5, *n* = 10) was similar to that observed for MCAO+OPNpt20 (500 ng) group (7.5 ± 0.7, *n* = 10); however, this value was significantly lower than that of the MCAO control group (Figure 3A). Although the mNSSs of the MCAO+OPNpt7RS (500 ng) group was markedly lower than that of the MCAO control group, it was slightly but significantly higher than that of the MCAO+OPNpt7R group (Figure 3A). However, no neurological improvement was detected in OPNpt7R-RAA-, OPNpt7R-RAD-, or OPNpt7R-sc-administered animals (Figure 3B). When motor activities were examined using a rotarod test at 5, 10, or 15 rpm at 2 days post-MCAO, the mean latency (time spent on the rod) in the MCAO+OPNpt7R group (500 ng, 1 h post-treatment) was significantly greater than that of the PBS-treated MCAO control group (Figure 3C). Increased mean latencies in the MCAO+OPNpt7R and MCAO+OPNpt20 groups were comparable at all rpms, and no improvement was detected in the MCAO+OPNpt7R-RAA group (Figure 3C). Similarly, in the grid walking test, marked improvement was also detected in the MCAO+OPNpt7R group but not in the OPNpt7R-RAA-treated group (Figure 3D). Taken together, these results indicate that improvements in sensory and motor functions are accompanied by the robust neuroprotective effects of OPNpt7R in the post-ischemic brain and that the RGD motif plays a critical role in this process. Changes in physiological variables were not detected in either the PBS- or the OPNpt7R-administered group when measured at 24 h post-MCAO (Table 1).

### 2.5. OPNpt7R Suppresses Activation of Microglial Cells and Augments M2 Polarization

To determine if OPNpt7R suppresses the activation of microglia, nitrite production was examined in primary microglia cultures after treating them with LPS in the presence or absence of OPNpt7R. When primary microglia cultures were co-treated with LPS (100 ng/mL) and various concentrations of OPNpt7R (0.01, 0.1, 0.5, or 1 µM) for 24 h, no effects were detected at any concentration used (Appendix A). However, when primary microglia cells were pre-incubated with 0.01 µM of OPNpt7R for 1 h and treated with LPS (100 ng/mL), nitrite production was reduced to 61.4 ± 6.0% (*n* = 4) of that in LPS-treated cells and similar levels of reduction were detected at 0.1, 0.5, or 1 µM of OPNpt7R (Figure 4A). When we compared the efficacy of OPNpt7R (0.1 µM) with that of OPNpt20 (0.1 µM) or OPNpt7RS (0.1 µM), the efficacy was comparable to that of OPNpt20 and significantly greater than that of OPNpt7RS (Figure 4B). Importantly, the same dose of OPNpt7Rsc failed to suppress nitrite production (Figure 4B), thus confirming the importance of the RGD motif in the OPNpt7R-mediated anti-inflammatory effect. Under all experimental conditions, cell viability was not altered (Appendix A). Furthermore, OPNpt7R significantly suppressed the LPS-induced upregulation of proinflammatory M1 marker (iNOS) (Figure 4C) and enhanced the induction of anti-inflammatory M2 markers (Arg1, CD206, and VEGF) (Figure 4D–F). However, OPNpt7R-sc neither suppressed M1 markers nor enhanced M2 markers (Figure 4C–F). When we compared anti-inflammatory efficacy of OPNpt7R to that of OPNpt20, it was comparable to OPNpt20 (Appendix A). Together, these results indicate that OPNpt7R augments M2 polarization, and that the RGD motif plays a critical role in this process.

### 2.6. Suppression of Inflammation and Induction of M2 Markers by OPNpt7R in the Post-Ischemic Brain 

We next investigated if OPNpt7R exhibits anti-inflammatory effects in the post-ischemic brain. Coronal brain sections that were obtained 1-day post-MCAO were stained with anti-Iba1 or anti-Mac-2 antibodies, markers for microglia/macrophages and activated microglia, respectively. In sham controls, Iba1+ cells were detected throughout the brain and exhibited a ramified morphology (Figure 5B). However, in PBS-treated MCAO controls, Iba1+ cells exhibited a phagocytic state in the cortical penumbras of ipsilateral hemispheres (black box in Figure 5A) 1-day post-MCAO (Figure 5C). In contrast, in the MCAO+OPNpt7R group, the majority of the Iba1+ cells retained a ramified morphology that was similar to that observed in sham controls (Figure 5D), while morphological changes of microglia to the phagocytic state were not suppressed in the MCAO+OPNpt7R-sc group (Figure 5E). For Mac-2, the number of Mac-2+ cells was markedly elevated in MCAO controls (Figure 5F,G); however, this number was decreased in the MCAO+OPNpt7R group and not in the MCAO+OPNpt7R-sc group (Figure 5H,I). The areas of Iba1^+^ or Mac-2^+^ cells in regions (0.5 mm^2^) indicated by the black box in A were measured and are presented (Figure 5J,K). Additionally, the induction of proinflammatory markers (IL-1β, IL-6, and TNF-α) observed at 1-day post-MCAO in the PBS-treated MCAO control group was significantly suppressed in the MCAO+OPNpt7R group (Figure 5L,M). Moreover, the induction of anti-inflammatory M2 markers (Arg1, IL-10, and IL-4) was enhanced in the MCAO+OPNpt7R group (Figure 5N,O). Neither the suppression of M1 markers nor the induction of M2 markers was detected in the MCAO+OPNpt7Rsc group (Figure 5L–O). Taken together, these results demonstrate the anti-inflammatory effects of OPNpt7R in the post-ischemic brain.

### 2.7. Enhancement of Phagocytic Activity of Microglia after Administration of OPNpt7R in the Post-Ischemic Brain

In addition to those M2 markers mentioned in a previous experiment (Figure 5N,O), we found that CD36, a scavenger receptor known as an M2 marker, was also significantly induced in OPNpt7R-administered MCAO animals (Figure 6A–C). The amount of CD36 was reduced in the post-ischemic brain (1 d post-MCAO), however, it was markedly recovered when OPNpt7R was administered intranasally at 1 h after MCAO (Figure 6A–C). These results prompted us to investigate if the phagocytosis of apoptotic cells by microglia is enhanced by OPNpt7R in the post-ischemic brain. To investigate the direct effect of OPNpt7R on the phagocytotic function of microglia, OPNpt7R (500 ng) was administered intranasally at 2 d post-MCAO, and phagocytic activity was visualized using a triple immunofluorescence labeling with antibodies against NeuN, Iba1, and activated caspase 3 at 3 d post-MCAO (Figure 6A). Compared to the microglia in sham-operated animals presenting a resting state, Iba1-positive cells showed an amoeboid shape, revealing an activation state at 3 d post-MCAO, and it was more evident in the ischemic core compared to the penumbra (Appendix A). Cleaved caspase-3 immunoreactivity was markedly enhanced at 3 d post-MCAO and it was mostly localized in neurons (NeuN-positive cells), indicating apoptotic death of neurons at 3 d post-MCAO (Appendix A). Quantification of phagocytized caspase 3-positive cells by microglia was carried out by the analysis of confocal images (Figure 6D–F). We defined phagocytized cells as those almost completely surrounded or completely encapsulated by the cell body or the extensions of the microglia. The ratio of phagocytosed caspase 3/neuN to the total number of caspase 3/neuN was presented as a phagocytic index and it was markedly increased in the OPNpt7R-administered MCAO group compared to the control MCAO and OPNpt7R-sc-administered group (Figure 6G). Together, these results indicated that the phagocytic capacity of microglia to remove apoptotic cells and cell debris was enhanced in OPNpt7R-administered MCAO animals and it might contribute to the inflammation resolution and neuroprotective effect of OPNpt7R in the ischemic brain.

## 3. Discussion

In the present study, we revealed that the OPN heptamer peptide (OPNpt7R) containing the RGD motif confers a robust neuroprotective effect in the post-ischemic brain and that it exhibits anti-inflammatory effects, suppresses the expression of M1 markers, and induces M2 polarization of microglia. Together with our previous report that demonstrated the enhancement of phagocytic efficacy of microglia by OPNpt7R [21], our present results indicate that augmentation of microglia M2 polarization may represent, at least in part, an important molecular mechanism underlying the neuroprotective effects of OPNpt7R in the post-ischemic brain.

In our previous study, we demonstrated that OPNpt20, an OPN icosamer peptide that harbors both the RGD and the SLAY motifs, exerts a robust neuroprotective effect [13] and proangiogenic effects in the post-ischemic brain [15]. In the present study, we revealed that the infarct suppressing effect of OPNpt7R, RGD-containing heptamer was comparable to that of OPNpt20. Although the importance of the RGD motif has been reported in studies using thrombin-cleaved OPN fragments for cell adhesion [22], and in 15 or 20 amino acid OPN peptides in animal models of Parkinson’s disease and stroke, respectively [2,13], we mapped the effective RGD-containing peptide to seven amino acids. We observed that the neuroprotective effect of OPNpt7R was significantly greater than that of OPNpt7RS, a heptamer containing both RGD and SLAY motifs (Figure 1 and Figure 4). These results are in agreement with our previous finding that both OPNpt7R and OPNpt7RS can enhance the motility and migration of BV2 cells, however, OPNpt7R exhibits slightly but significantly higher potencies than OPNpt7RS [21]. However, as improvements in cardiac function and angiogenic activity have been observed in SVVYGLR peptide-treated animal models of myocardial fibrosis and heart failure [23,24,25,26], we cannot exclude the possibility of differential effects of the RGD and SLAY motifs in different contexts. Additionally, in OPNpt7RS, the close proximity of RGD and SLAY motifs might allow only one motif to be occupied by the corresponding integrin, and competitions among integrins for the binding with adjacent binding sites may lower the efficacy of this peptide. This explanation requires further study for confirmation.

Induction of OPN occurs as early as 24 h and peaks at 3 or 5 days after the ischemic insult, respectively, in transient or permanent focal ischemia rat models [27,28]. In our previous reports, we demonstrated that exogenously administered recombinant OPN or OPN icosamer peptide (OPNpt20) before the induction of endogenous OPN (e.g., 1 h after MCAO) confers a robust neuroprotective effect in the post-ischemic brain [11,13]. Here, we observed that the neuroprotective effects of OPNpt7R administered 6 or 1 h prior to MCAO were similar to that of 1 h post-treatment and interestingly, 6 h post-treatment also significantly reduced infarct volume, albeit at relatively low efficacy (Figure 2C,D). These results indicate that OPNpt7R exerts beneficial effects during the acute and subacute phases by accelerating the time points of the beneficial effects provided by the endogenous OPN. Regarding the specific effects, given that the beneficial effects of OPN or OPN-derived peptides reported in the brain are pleiotropic, for example, anti-excitotoxic [29,30], anti-inflammatory [11,13], pro-angiogenic [15,31,32], and phagocytosis-enhancing [14,19,21] ones, it can be speculated that exogenous OPNpt7R performs differential functions depending on the administration time points. Further studies are necessary to identify any differential functions of OPNpt7R when administered at different time points.

Microglia are highly plastic cells that engage in different functions under different environmental conditions by adjusting their morphology and functions depending on the specific spatiotemporal cues [33]. In pathological conditions such as cerebral ischemia, microglia are rapidly activated and produce numerous pro- or anti-inflammatory effector molecules depending on their activation state (M1 or M2) to exacerbate damage or protect the brain from injury [16]. Augmentation of microglial M2 polarization by OPN has been reported in permanent focal cerebral ischemia [18,34], and OPN-induced attenuation of secondary neurodegeneration in the thalamus after experimental stroke has also been reported recently in a photothrombotic stroke model [35]. Here, we found that OPNpt7R possesses the potential to polarize microglia to M2, upregulating numerous M2 markers such as Arg1, CD206, and VEGF (Figure 4D–F). In particular, Arg1, which is the functional antagonist of iNOS, and VEGF, which has been shown to polarize microglia to M2 in a rat model of focal cerebral ischemia [36], were both significantly upregulated (Figure 5N,O). Although the M1/M2 dichotomy may not be absolute and the distinction between the two subpopulations was not that strict, we observed that OPNpt7R strongly biased biomarker expression toward the M2 type both in vitro and in vivo.

Among various functions of M2 microglia, phagocytosis has been considered to confer beneficial effects on repair and regeneration. In numerous neurological diseases, activated microglia are highly mobile, migrate to damaged regions, and remove damaged neurons and their cell debris through phagocytosis. The importance of microglia phagocytosis has been reported not only in neurodegenerative diseases such as Alzheimer’s disease, Parkinson’s disease, amyotrophic lateral sclerosis, and multiple sclerosis [37] but in the postischemic brain, in which the phagocytosis of dead neurons is crucial as it promotes axon regeneration and restoration of the brain microenvironment [37]. In our previous report, we showed that phagocytic activity and F-actin polymerization are significantly enhanced in OPNpt7R-treated BV2 cells in an RGD-dependent manner and the EERK, FAK, and AKT signaling pathways are involved in the induction of phagocytic activity by OPNpt7R [21]. Here, we confirmed the OPNpt7R-induced phagocytic activity of microglia in the post-ischemic brain (Figure 6). Similar activity was observed, albeit for full OPN protein, in a mouse model of Alzheimer’s disease [20] and in obesity-associated adipose tissue [38], wherein phagocytosis induction by OPN was accompanied by macrophage polarization toward an anti-inflammatory phenotype, implicating that modulation of microglial phagocytosis by OPN offers a potential therapeutic strategy for treating neurological diseases. Therefore, it can be speculated that in the post-ischemic brain, increased extracellular OPNpt7R exerts a robust neuroprotective function by providing an environmental cue to microglia to initiate autocrine signaling changes and promote paracrine beneficial effects via interactions with other cell types.

## 4. Materials and Methods

### 4.1. Peptides

Eight OPN peptides, RGDSLAY-containing 20-, 13-, 7-amino acid peptides (OPNpt20, OPNpt13, OPNpt7RS), RGD-containing 7-amino acid peptide (OPNpt7R), SLAY-containing 7-amino acid peptide (OPNpt7S), and three mutant heptamers, OPNpt7-RAA (RGD was replaced with RAA), OPNpt7-RAD (RGD was replaced with RAD) or OPNpt7-sc (scrambled peptide), were synthesized (PEPTRON, Daejeon, South Korea). OPNpt20 was used as a positive control and OPNpt7-sc was used as a negative control.

### 4.2. Surgical Procedure Used for Middle Cerebral Artery Occlusion

Male Sprague-Dawley rats (8–9 weeks of age) were housed under diurnal lighting conditions and provided with food and tap water ad libitum. All animal studies were performed out in strict accordance with the Guide for the Care and Use of Laboratory Animals published by the National Institute of Health (NIH, USA 2013) and in accordance with ARRIVE guidelines (http://www.nc3rs.org/ARRIVE (accessed on 31 August 2021)). The animal protocol used in this study was reviewed and approved by the INHA University Institutional Animal Care and Use Committee (INHA-IACUC) in regard to ethical standards (Approval Number INHA-180105-531-2). MCAO was performed as previously described [39]. Briefly, rats (250–300 g) were anesthetized with 5% isoflurane in a 30% oxygen/70% nitrous oxide gas mixture. Anesthesia was maintained during procedures using 0.5% isoflurane in the same gas mixture, and occlusion of the right middle carotid artery was induced for 1 h by advancing a nylon suture (4-0; AILEE, Busan, South Korea) with a heat-induced bulb at its tip (∼0.3 mm in diameter) along the internal carotid artery for a distance of 20–22 mm from the bifurcation of the external carotid artery. Reperfusion was allowed for up to 3 days. A thermoregulated heating pad and a heating lamp were both used to maintain a rectal temperature of 37 ± 0.5 °C during surgery. Animals were randomly allocated to a sham (*n* = 20), MCAO (*n* = 45), MCAO+OPNpt20-treated (*n* = 21), MCAO+OPNpt13R-treated (*n* = 10), MCAO+OPNpt7R-treated (*n* = 53), MCAO+OPNpt7RS-treated (*n* = 10), MCAO+OPNpt7S-treated (*n* = 4), MCAO+OPNpt7RAA-treated (*n* = 7), MCAO+OPNpt7RAD-treated (*n* = 4), or MCAO+OPNpt7R-sc-treated (*n* = 28) group. Animals assigned to the sham group underwent an identical procedure; however, the MCA was not occluded. All measurements, including infarct volumes, neurological deficits, and latency in a rotarod test were conducted in a blind manner.

### 4.3. Intranasal Administration

Intranasal administration was performed as previously described by Kim et al. (2012) [40]. Briefly, rats were anesthetized with an intramuscular injection of a mixture of ketamine (3.75 mg/100 g body weight) and xylazine hydrochloride (0.5 mg/100 g per body weight). A nose drop containing 500 ng of OPN peptide dissolved in PBS (20 μL) was carefully placed in each nostril of anesthetized animals (supine 90° angle) using a pre-autoclaved pipet tip (T-200-Y, Axygen, CA, USA). The procedure was repeated until the entire dosage was administered at 2 min intervals between applications.

### 4.4. Infarct Volume Assessment

Rats were decapitated at 2 days post-MCAO and whole brains were dissected coronally into 2 mm slices using a metallic brain matrix (RBM-40000, ASI, Springville, UT, USA). Slices were immediately stained by immersing them in 2% 2,3,5-triphenyl tetrazolium chloride (TTC) at 37 °C for 15 min and then fixed in 4% paraformaldehyde (PFA). Infarcted tissue areas were measured using the Scion Image program (Scion Corporation, Frederick, MD, USA). To account for edema and shrinkage, areas of ischemic lesions were calculated using the formula: contralateral hemisphere volume—(ipsilateral hemisphere volume—measured injury volume). Infarct volumes were quantified (in mm^3^) by summing infarct sizes on adjacent tissue sections.

### 4.5. Modified Neurological Severity Scores (mNSS)

Neurological deficits were evaluated using mNSSs at 2 days post-MCAO. The mNSS system consists of motor, sensory, balance, and reflex tests, and the overall results are graded according to a scale of 0 to 18 (normal: 0, maximal deficit: 18) [41]. Motor scores were determined: (1) by suspending a rat by its tail and awarding a score of zero or one (total score 0–3) for forelimb flexion, hindlimb flexion, and head movement by >10° with respect to the vertical axis within 30 s and (2) by placing a rat on the floor and awarding scores from 0 to 3 for normal walking (0), inability to walk straight (1), circling toward the paretic side (2), or falling on the paretic side (3). Sensory tests included a placing test (score 0–1) and a proprioceptive test (score 0–1). The beam balance test was used to test balance, and scores ranging from 0 to 6 were allocated based on balancing with a steady posture (0), grasping the side of the beam (1), hugging the beam with one limb off the beam (2), hugging the beam with two limbs off the beam or rotating around the beam for over 60 s (3), attempting to balance on the beam but falling off within 20 to 40 s (4), attempting to balance on the beam but falling off within 20 s (5), or making no attempt to balance or hang onto the beam (6). Reflex was scored based on the following four items (maximum possible score of 4): pinna reflex (0–1), corneal reflex (0–1), startle reflex (0–1), seizures, myoclonus, or myodystony (0–1).

### 4.6. Rotarod Test

Twenty-four hours prior to MCAO, rats were conditioned on a rotarod unit at a constant 3 rpm speed until they were able to remain on the rotating spindle for 180 s. At 2 days post-MCAO, rats were subjected to a rotarod test at spindle speeds of 5, 10, or 15 rpm, and residence times on the spindle were recorded. A 1 h rest period was allowed between tests.

### 4.7. Grid Walking Test

Twenty-four hours prior to MCAO, rats were conditioned to cross a horizontal ladder. At 2 days post-MCAO, each animal was placed on the grid, and the number of foot-fault errors was monitored and recorded until the rats crossed the horizontal ladder.

### 4.8. Primary Microglia Culture

Primary microglial cultures were prepared as previously described [39]. Briefly, cells that dissociated from the cerebral hemispheres of 1-day-old postnatal rat brains (Sprague-Dawley strain) were seeded at a density of 1.2 × 10^6^ cells/mL in Dulbecco’s modified Eagle’s medium (DMEM; Gibco, Carlsbad, CA, USA) containing 10% FBS (Hyclone, Logan, UT, USA) and 1% penicillin-streptomycin (Gibco BRL, Gaithersburg, MD, USA). After 2 weeks, microglia were detached from the flasks by mild shaking and filtered through a cell strainer (BD Falcon, Bedford, MA, USA) to remove astrocytes. After centrifugation (1000× *g*) for 5 min, the cells were resuspended in fresh DMEM containing 10% FBS and 1% penicillin-streptomycin and then plated at a final density of 1.5 × 10^5^ cells/well on a 24 multi-well culture plate. After 2 h, the medium was changed to DMEM containing 5% FBS and 500 µM B27 supplement (Gibco BRL, Gaithersburg, MD, USA).

### 4.9. Nitrite Measurements

Primary microglia cells (1.5 × 10^5^) were seeded into 24-well plates and treated with LPS (100 ng/mL) after 1 day. To measure the amount of NO produced, 50 µL of conditioned medium was mixed with an equal volume of Griess reagent (0.5% sulfanilamide, 0.05% N-naphthylene-diamine-H-chloride, and 2.5% H_3_PO_4_) and incubated for 5 min at room temperature. The absorbance of the mixtures was measured at 550 nm using a microplate reader. NaNO_2_ standards were used to calculate the NO_2_^−^ concentrations.

### 4.10. Immunoblot Analysis

Cells were washed twice with cold PBS and lysed in RIPA buffer (50 mM Tris-HCl [pH 7.4], 1% NP-40, 0.25% sodium deoxycholate, 150 mM NaCl, and complete Mini protease inhibitor cocktail tablets [Roche, Basel, Switzerland]). Lysates were then centrifuged at 15,870× *g* at 4 °C for 10 min, and the supernatants were loaded onto 12% SDS PAGE gels. The resolved proteins were transferred onto polyvinylidene fluoride membranes (BioRad, Hercules, CA, USA), and the blots were incubated with the antibodies overnight at 4 °C. Primary antibodies were diluted 1:1000 for anti-iNOS, Arg1, CD206 (Cell Signaling, Beverly, MA, USA), anti-VEGF (Abcam, Cambridge, UK), anti-CD36 (Bioss, Woburn, MA, USA), and α-tubulin (Santa Cruz Biotechnology, Santa Cruz, CA, USA). The next day, membranes were detected using a chemiluminescence kit (Amersham, Arlington Heights, IL, USA) and anti-rabbit and anti-mouse HRP-conjugated secondary antibodies (1:2000, Santa Cruz Biotechnology, Santa Cruz, CA, USA).

### 4.11. Immunohistochemistry

Animals were sacrificed 1-day post-MCAO, and their brains were fixed in 4% paraformaldehyde by transcardiac perfusion and then post-fixed in the same solution overnight at 4 °C. Brain sections (20 μm) were produced using a vibratome, and immunological staining was performed as previously described [42]. Primary antibodies were incubated after diluting to 1:500 for anti-ionized calcium-binding adaptor molecule-1 (Iba1; Wako Pure Chemicals, Osaka, Japan) and to 1:250 for anti-Mac-2 (Abcam, Cambridge, UK). After washing with PBS containing 0.1% Triton X-100, the sections were incubated with anti-mouse IgG (Vector Laboratories, Burlingame, CA, USA) for anti-Mac-2 or anti-rabbit IgG (Vector Laboratories, Burlingame, CA, USA) for anti-Iba1 in PBS at room temperature for 1 h and visualized using the HRP/3,3′-diaminobenzidine (DAB) system (Vector Laboratories, Burlingame, CA, USA).

### 4.12. Immunofluorescence Staining

The fixed brain tissues were sectioned as previously described in immunohistochemistry. Primary antibodies were diluted to 1:200 for anti-ionized calcium-binding adaptor molecule-1 (Iba1) (Wako Pure Chemicals, Osaka, Japan), for anti-neuronal nuclei (NeuN) (Merck Millipore Corporation, Darmstadt, Germany), and for anti-cleaved caspase-3 (Cell signaling, Danvers, MA, USA) antibodies. Brain tissues were soaked in diluted primary antibody solutions overnight at 4 °C and then washed with PBS three times. Sections were incubated with 1:300 of Alexa Fluor 488-conjugated anti-goat IgG (Thermo Fisher Scientific, Waltham, MA, USA) for anti-Iba1, 1:300 of rhodamine-labeled anti-mouse IgG (Merck Millipore Corporation, Darmstadt, Germany) for anti-NeuN, or 1:200 of Alexa Fluor 633-conjugated anti-rabbit IgG (Thermo Fisher Scientific, Waltham, MA, USA) for anti-cleaved caspase-3 in PBS for 1 h at room temperature. The sections were then mounted on slides with a VECTASHIELD Antifade Mounting solution containing DAPI (Vector Laboratories, Burlingame, CA, USA) and examined under a Zeiss LSM 510 META microscope (Carl-Zeiss-Strasse, Oberkochen, Germany).

### 4.13. RNA Preparation and Reverse Transcription Polymerase Chain Reaction (RT-PCR)

The 6-mm coronal brain slices, 6 to 12 mm apart starting from the frontal pole of the frontal cortex, were prepared by a brain matrix device (RBM-40000, ASI Instruments, Houston, TX, USA). Total RNA was prepared using a TRIzol reagent (Thermo Fisher Scientific, Waltham, MA, USA) and 1 µg RNA samples were used for cDNA synthesis using an RT-PCR kit (Roche, Mannheim, Germany). Sequences for primers sets were summarized in Table 2.

### 4.14. Statistical Analysis

Two-sample comparisons were performed using the Student’s *t*-test, and multiple comparisons were achieved using one-way or two-way analysis of variance (ANOVA) followed by Tukey’s post hoc test. PRISM software 5.0 (Graph Pad Software) was used for the analysis. The results are presented as the mean ± SEM, and statistical differences were indicated by *p* values < 0.05.

## Figures and Tables

**Figure 1 ijms-22-09999-f001:**
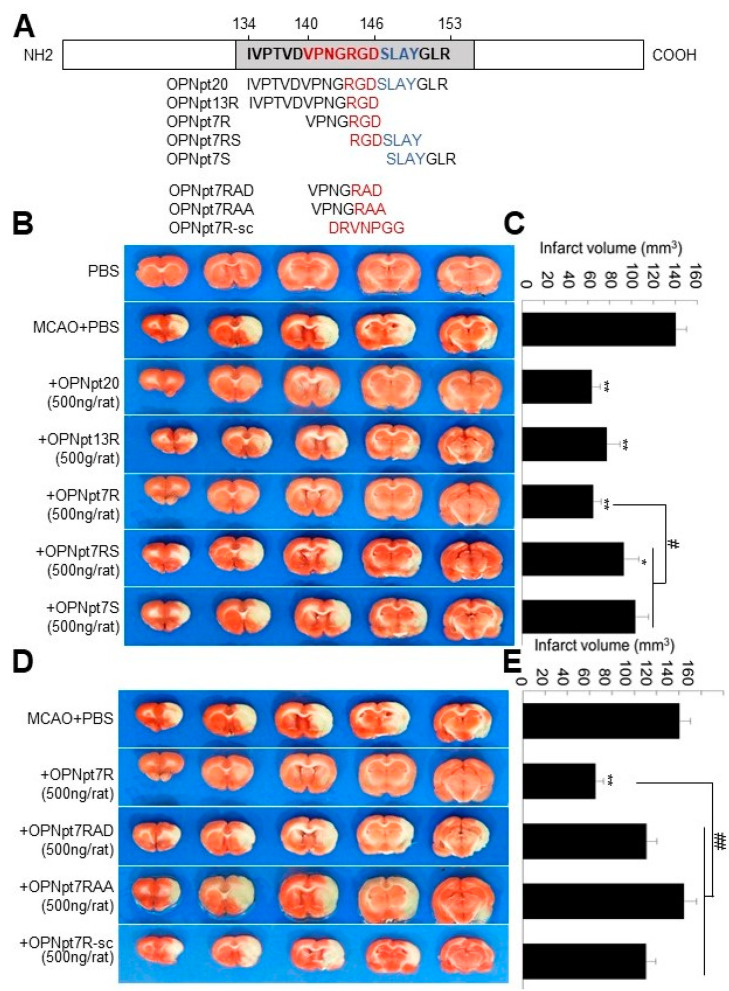
Reduction in infarct volume by intranasally administered OPNpt7R and the importance of the RGD motif; (**A**) schematic diagram presenting the amino acid sequences of OPNpt20, OPNpt13R, OPNpt7R, OPNpt7RS, OPNpt7S, OPNpt7R-RAD, OPNpt7R-RAA, and OPNpt7R-sc. (**B**,**C**) OPNpt20 (500 ng), OPNpt13R (500 ng), OPNpt7R (500 ng), OPNpt7RS (500 ng), or OPNpt7S (500 ng) was administered intranasally at 1 h post-MCAO (60 min), and TTC staining was performed at 2 days post-MCAO. (**D**,**E**) OPNpt7R (500 ng), OPNpt7R-RAD (500 ng), OPNpt7R-RAA (500 ng), or OPNpt7R-sc (500 ng) was administered intranasally at 1 h post-MCAO, and TTC staining was performed at 2 days post-MCAO. Representative images of infarctions are shown (**B**,**D**), and mean infarction volumes are presented as the mean ± SEM (*n* = 4–11) (**C**,**E**). Sham, sham-operated rats; MCAO, PBS-treated MCAO control; OPNpt20, OPNpt20-treated MCAO rats; OPNpt13R, OPNpt13R-treated MCAO rats OPNpt7R, OPNpt7R-treated MCAO rats, OPNpt7RS, OPNpt7RS-treated MCAO rats, OPNpt7S, OPNpt7S-treated MCAO rats, OPNpt7R-RAD, OPNpt7R-RAD-treated MCAO rats, OPNpt7R-RAA, OPNpt7R-RAA-treated MCAO rats, OPNpt7R-sc, scrambled OPNpt7R-treated MCAO rats. * *p* < 0.05, ** *p* < 0.01 vs. PBS-treated MCAO control, # *p* < 0.05, ## *p* < 0.01 between indicated groups.

**Figure 2 ijms-22-09999-f002:**
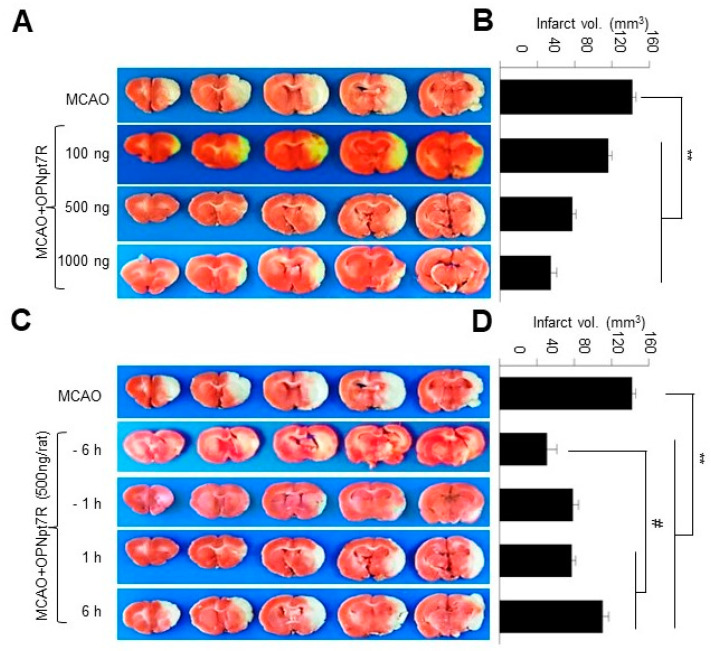
Reduction in infarct volume by intranasally administered OPNpt7R at various concentrations and times; OPNpt7R (100, 500, or 1000 ng) was administered intranasally at 1 h post-MCAO (**A**,**B**) and OPNpt7R (500 ng) was administered intranasally at 6 or 1 h prior to- or 1 or 6 h post-MCAO (60 min) (**C**,**D**). TTC staining was performed at 2 days post-MCAO. Representative images of infarctions are shown (**A**,**C**) and mean infarction volumes are presented as the mean ± SEM (**B**,**D**). Sham, sham-operated rats (*n* = 3); MCAO, PBS-treated MCAO control (*n* = 4); OPNpt7R, OPNpt7R-treated MCAO rats (*n* = 28). ** *p* < 0.01 vs. PBS-treated MCAO control, # *p* < 0.05, between indicated groups.

**Figure 3 ijms-22-09999-f003:**
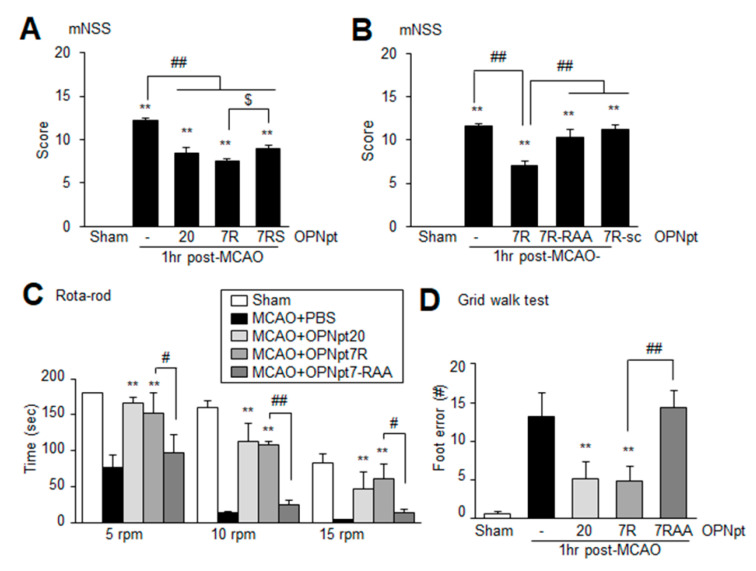
Recovery of neurological and motor deficits of MCAO rats by OPNpt7R; (**A**,**B**) OPNpt20 (500 ng), OPNpt7R (500 ng), OPNpt7RS (500 ng), OPNpt7R-RAA (500 ng), OPNpt7R-RAD (500 ng), or OPNpt7R-sc (500 ng) was administered intranasally at 1 h post-MCAO (**A**), and neurological deficits were evaluated using modified neurological severity scores at 2 days post-MCAO. (**C**,**D**) OPNpt20 (500 ng), OPNpt7R (500 ng), or OPNpt7R-RAA (500 ng) was administered intranasally 1 h post-MCAO, and the rota-rod test (**C**) and grid walk test (**D**) were both performed 2 days post-MCAO. The rotarod test was performed at 5, 10, and 15 rpm at the indicated times with an 1 h inter-trial rest period. Data are presented as the mean ± SEM. Sham, sham-operated rats (*n* = 4); MCAO, PBS-treated MCAO controls (*n* = 10); MCAO+OPNpt20, OPNpt20-treated MCAO rats (*n* = 10); MCAO+OPNpt7R, OPNpt7R-treated MCAO rats (*n* = 10); MCAO+OPNpt7RS, OPNpt7RS-treated MCAO rats (*n* = 4); MCAO+OPNpt7R-RAA, OPNpt7R-RAA-treated MCAO rats (*n* = 3); MCAO+OPNpt7R-sc, OPNpt7R-sc-treated MCAO rats (*n* = 4). ** *p* < 0.01 vs. PBS-treated MCAO controls, # *p* < 0.05, ## *p* < 0.01, $ *p* < 0.05 between indicated groups.

**Figure 4 ijms-22-09999-f004:**
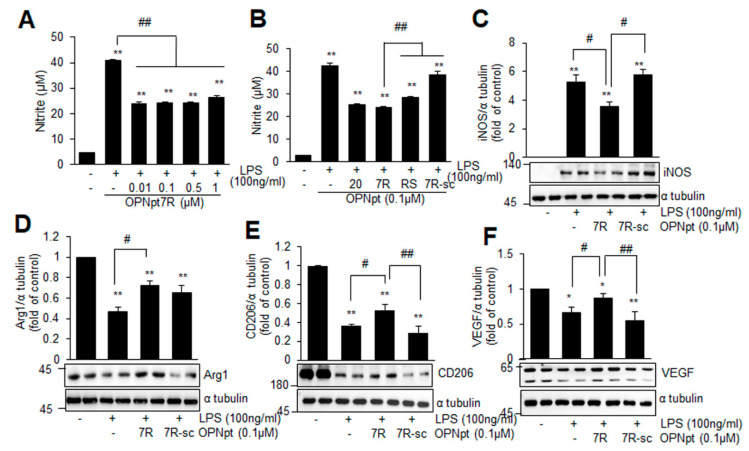
Anti-inflammatory effect of OPNpt7R in primary microglial cultures; (**A**,**B**) primary microglial cultures were pre-incubated with OPNpt7R (0.01, 0.1, 0.5 or 1 μM) for 1 h (**A**) or with 0.1 μM of OPNpt20, OPNpt7R, OPNpt7RS, or OPNpt7R-sc for 1 h (**B**) and treated with LPS (100 ng/mL) for 24 h. Nitrite production was measured using the Griess assay. (**C**,**D**) Primary microglial cultures were pre-incubated with 0.1 μM of OPNpt7R or OPNpt7R-sc for 1 h and treated with LPS (100 ng/mL). Levels of proinflammatory (iNOS) (**C**) or anti-inflammatory (Arg1, CD206, and VEGF) (**D**–**F**) markers were determined by immunoblotting, and the results are presented as the mean ± SEM (*n* = 3–6). * *p* < 0.05, ** *p* < 0.01 versus LPS-treated cells, # *p* < 0.05, ## *p* < 0.01 between indicated groups.

**Figure 5 ijms-22-09999-f005:**
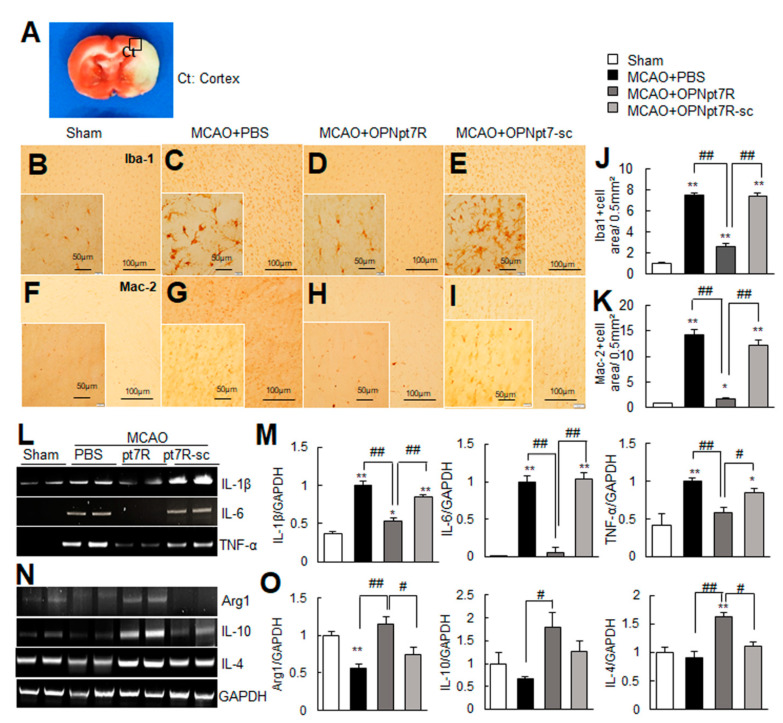
Anti-inflammatory effect of OPNpt7R in the post-MCAO brain; OPNpt7R (500 ng) or OPNpt7R-sc (500 ng) was administered intranasally 1 h post-MCAO. (**A**–**K**) Coronal brain sections were prepared from sham (**B**,**F**), MCAO (**C**,**G**), MCAO+OPNpt7R (**D**,**H**), and MCAO+OPNpt7R-sc (**E**,**I**) groups at 1 day after MCAO. Brain sections were processed for immunostaining with anti-Iba1 (**B**–**E**) or anti-Mac-2 (**F**–**I**) antibodies. Representative pictures from three independent experiments are presented (**B**–**E**,**F**–**I**), and the areas of Iba1^+^ or Mac-2^+^ cells in regions (0.5 mm^2^) indicated black box in A were measured and are presented as means ± SEMs (*n* = 12, 12 brain slices from three animals) (**J**,**K**). The insets are high-magnification photographs of each image. Scale bars in B-I represent 100 μm, and those in the insets represent 50 μm. (**L**–**O**) Samples were prepared from the indicated region in A at 1-day post-MCAO, and expression levels of pro- or anti-inflammatory markers were examined by RT-PCR. The results are presented as the mean ± SEM (*n* = 3~4). Sham, sham-operated rats; MCAO, PBS-treated MCAO controls; MCAO+OPNpt7R, OPNpt7R-treated MCAO rats; MCAO+OPNpt7R-sc, OPNpt7R-sc-treated MCAO rats. * *p* < 0.05, ** *p* < 0.01 vs. PBS-treated MCAO controls, and # *p* < 0.05, ## *p* < 0.01 between indicated groups.

**Figure 6 ijms-22-09999-f006:**
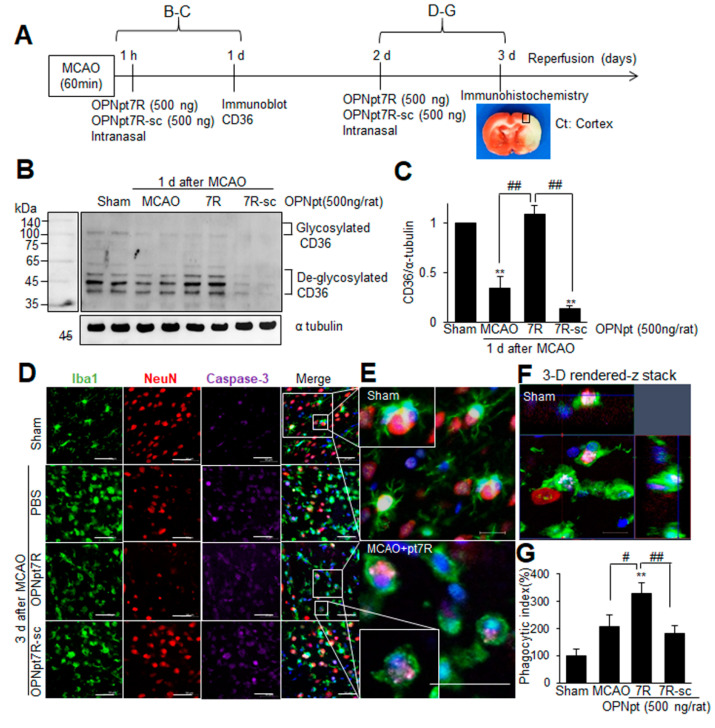
Enhanced phagocytic activity of microglia by OPNpt7R in the post-MCAO brain; (**A**) OPNpt7R (500 ng) or OPNpt7R-sc (500 ng) was administered intranasally at 1 h post-MCAO (**B**,**C**) or at 2 d post-MCAO (**D**–**G**). (**B**,**C**) Protein samples were prepared from the indicated region in A at 1 d post-MCAO, and the CD36 level was examined by immunoblot analysis. Representative images are presented (**B**) and quantified results are presented as the mean ± SEM (*n* = 3) (**C**). (**D**–**G**) Coronal brain sections were prepared from sham, MCAO, MCAO+OPNpt7R, and MCAO+OPNpt7R-sc groups at 3 d after MCAO and processed for immunostaining with anti-Iba1, anti-NeuN, and anti-activated caspase 3 antibodies. Representative pictures are presented (**D**–**F**) and phagocytic index (the ratio of Caspase 3-encapsulated in Iba1 over Caspase 3 in regions (0.2 mm^2^) indicated as black box in A are presented as the mean ± SEM (*n* = 10, 10 brain slices from three animals) (**G**). The images in E are high-magnification photographs of images in the Sham and MCAO+OPNpt7R group (indicated as white boxes). Scale bars represent 50 μm. Sham, sham-operated rats; MCAO, PBS-treated MCAO controls; MCAO+OPNpt7R, OPNpt7R-treated MCAO rats; MCAO+OPNpt7R-sc, OPNpt7R-sc-treated MCAO rats. ** *p* < 0.01 vs. PBS-treated MCAO controls, and # *p* < 0.05, *## p* < 0.01 between indicated groups.

**Table 1 ijms-22-09999-t001:** Physiological parameters.

	PBS-Treated Group		OPNpt7R-Treated Group	
	Base	During Ischemia	Base	During Ischemia
pH	7.6 ± 0.1	7.6 ± 0.2	7.5 ± 0.2	7.5 ± 0.7
pO_2,_ mmHg	146.8 ± 2.1	151 ± 4.0	144.6 ± 2.1	146.2 ± 6.4
pCO_2,_ mmHg	35.9 ± 0.5	33.2 ± 2.4	35.2 ± 2.2	33.9 ± 2.6
Glucose, mg/dL	109.4 ± 3.5	100.6 ± 5.1	114 ± 5.7	97.6 ± 7.8
Temperature	36.6 ± 0.7	37.8 ± 1.7	36.5 ± 2.5	37.5 ± 0.6

**Table 2 ijms-22-09999-t002:** Oligonucleotide primers used for the RT-PCR analysis.

Gene (GenBank Accession No.)	Oligonucleotide Primer Sequences	PCR Product Size (bp)	Tm	Cycle
IL-1β (M98820)	5′-AGC ATC CAG CTT CAA ATC TCA-3′ 5′-CGA GGC ATT TTT GTT GTT CAT-3′	268	54	25
IL-6 (M26744)	5′-CAA GAG ACT TCC AGC GAG TTG-3′ 5′-GAA ACG GAA CTC CAG AAG ACC-3′	350	54	25
TNF-α (NM012675)	5′-CTC CGT GAT GTC TAA GTA CT-3′ 5′-CTC AAA ACT CGA GTG ACA AG-3′	422	54	25
IL-10 (XM006249712)	5′-CTT TCA CTT GCC CTC ATC-3′ 5′-ACA AAC AAT ACG CCA TTC-3′	265	47	30
Arg1 (AH002138)	5′-CAG AAG AAT GGA AGA GTC AG-3′ 5′-CAG ATA TGC AGG GAG TCA CC-3′	524	54	30
IL-4 (AY496861)	5′-ACC TTG GTG TCA CCC TGT TCT GC-3′ 5′-GTT GTG AGC GTG GAC TCA TTC ACG-3′	292	55	30
GAPDH (DQ403053)	5′-TCATTGACCTCAACTACATGGT-3′ 5′-CTAAGCAGTTGGTGGTGCAG -3′	363	55	25

## Data Availability

The data presented in this study are available on request from the corresponding author.

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
