# Peer review of "Intranasal Delivery of RGD-Containing Osteopontin Heptamer Peptide Confers Neuroprotection in the Ischemic Brain and Augments Microglia M2 Polarization"

_ijms, 2021, doi:10.3390/ijms22189999_

Round 1

Reviewer 1 Report

In this manuscript, the authors investigated whether the motif containing RGD play a role in neuroprotective effect and whether OPNpt7R play a role in the M2 polarization of microglia and confers neuroprotection in ischemic brain. The authors did a detailed tests at the animal level. However, some descriptions need to be further ensured. The major revision should be completed for publication. Some personal comments are listed below:This work is an extension of previous reviews. Some conclusion and description are similar to “Yin-Chuan Ji et al. Proangiogenic functions of an RGD-SLAY-containing osteopontin icosamer peptide in HUVECs and in the postischemic brain[J]. Experimental and Molecular Medicine, 2018, 50(S10): 1-10”.Some grammar errors have been observed in this paper.Many format errors still exist. For example, the authors incorrectly wrote ng as mg and Fig as Figs in section 2. Please check the whole manuscript and polish the writing again.

Author Response

In this manuscript, the authors investigated whether the motif containing RGD play a role in neuroprotective effect and whether OPNpt7R play a role in the M2 polarization of microglia and confers neuroprotection in ischemic brain. The authors did a detailed tests at the animal level. However, some descriptions need to be further ensured. The major revision should be completed for publication. Some personal comments are listed below:

  1. This work is an extension of previous reviews. Some conclusion and description are similar to “Yin-Chuan Ji et al. Proangiogenic functions of an RGD-SLAY-containing osteopontin icosamer peptide in HUVECs and in the postischemic brain[J]. Experimental and Molecular Medicine, 2018, 50(S10): 1-10”.

Response: Thank you for raising this point. OPN contains RGD and SLAY motifs, which bind to several integrins, mediating a wide range of cellular process. The RGD and SLAY motifs interact with different integrins and perform different functions, but there is still insufficient research as to which motif is more important when alone. In the present study, neuroprotective effect of OPN-peptide heptamer containing RGD, SLAY, or both of RGD and SLAY was investigated in primary microglia and in the postischemic brain. By following reviewer’s comment, we checked the whole manuscript, focusing and discussing this point.

  1. Some grammar errors have been observed in this paper. Many format errors still exist. For example, the authors incorrectly wrote ng as mg and Fig as Figs in section 2. Please check the whole manuscript and polish the writing again.

Response: By following reviewer’s comment, we changed all “Figs to Fig” and “1 mg to 1000 ng” in section 2. In addition, we checked the whole manuscript with great care.

Reviewer 2 Report

The authors should correct the missing details on the methods and add experimental controls to achieve reproducibility and improve the paper quality.

Line 10:  various tissues, including the brain. Suggestion: change the word ‘tissue’ to the word ‘organ.’ The brain has several different tissues, such as connective and nervous tissue.

Several lines:  I believe that what you are measuring is efficacy and not potency. “Potency refers to the concentration (EC50) or dose (ED50) of a drug required to produce 50% of that drug’s maximal effect”. “For clinical use, it is important to distinguish between a drug’s potency and its efficacy. The clinical effectiveness of a drug depends not on its potency (EC50), but on its maximal efficacy and its ability to reach the relevant receptors”.

Line 57: add:  For review, see 17 (it is a review).

Line 62: Please add a reference for the role of OPN in the phagocytic function of monocytes.

What is the rationale for 48 hours? I believe that is because of the time of microglia polarization, even though it is not the day where the maximum expression of the markers occurs.  Is it? If I am correct, this information is not on the paper. Please add the rationale for the 48 hours timepoint.

Line 78, results: What is the motive to have many different n (number of animals) for the MCAO and the peptides? Did the authors perform a power analysis? 

Line 108: Why have the authors used a post hoc SNK test? Please use Tukey’s post hoc test because it is more powerful than the SNK test. Maybe your results could change. “Because the Newman-Keuls test works sequentially, it cannot produce 95% confidence intervals for each difference or multiplicity-adjusted exact P values.  In contrast, the Tukey test can compute both confidence intervals and adjusted P values”. Also, did the authors removed the outliers or not? Please add this information.

Line 110: The authors do not measure potency; they measure efficacy.

Surgical procedure: It is the most crucial weakness. Why did the authors not verify the decrease of cerebral blood flow using laser doppler? Did the authors induce reductions of more than 70% in all studied animals?

Discussion: The authors are just discussing figs 1, 4 2C, and 2D. The discussion for the rest of the figures should be included as well.

Methods: There is no information about the RT-PCR cycles in the thermocycler, or if the samples were treated with DNAse, or if the primers were in the same exon (which is why the DNAse treatment is necessary). How do the authors ensure that the amplification corresponds to an mRNA-cDNA amplification and not to DNA contamination? The differences could be higher if the authors use Real-time PCR or at least if they decrease the PCR cycles to a logarithmic phase of amplification.

Figure 4: Pay attention to the legend of the Y-axis. They are superposed on A and C. iNOS, not Inos.

-The scrambled peptide 7R-sc has the same effect as 7R in the Arg1 protein expression. Why? This is not a reasonable control for this experiment.

Figure 4.9. Did the conditioned media contain Phenol Red? Was the standard curve prepared using the same medium?

Figure 5: It is impossible to analyze Arg1 with PBS. First, there is a smear, and second, the amplicons do not have the correct size. This smear could appear when the amount of mRNA is high. Then, it is hard to believe in these results as they are presented. Strangely, what you considered negative, could be positive; there is no curve of concentration. As the authors are not using Real-time PCR, at least two controls should be shown: 1)  a complete agarose gel with a DNA ladder; 2) negative and positive controls treated with DNAse or without reverse transcriptase.

-Please explain why the intensity of the Iba staining is different.

-In the comparative picture of figure 5. From the image, I have to disagree that figure D has fewer cells than figure C.

-Please provide the negative controls of the secondary alone in Figure 5. Also, please provide a detailed description of the analysis. A counterstaining will be of great help to verify if this region has other cells or it is a region analyzed contain a more infarcted area in the cortex.

 How many people analyze the videos of mNSS? Was it a double-blind experiment? Please add.

Line 460: What is the pore size of the PVDF membrane? This information will help to reproduce the results.

-How much of the protein extract was injected? Did the authors make an antibody titration or change the protein concentration to avoid saturation? It is different from chemiluminescence saturation.

Reviewer 3 Report

In this study the authors investigated the contribution of RGD motif in the neuroprotective and anti-inflammatory properties of osteopontin (OPN) in MCAO. They tested several OPN peptides of various length containing RGD, SLAY and scrambled motifs both in vivo and in vitro. They found that RGD has a key role in neuroprotective effects of OPN. The study is generally well done and well written. There are only minor points to address:

  1. Panels B, D in Fig 1 and A, C in Fig 2 should be larger as it is hard to see the infarct volume.
  2. Results section 2.6: In the last sentence a N-NAM is mentioned. Is this an error?
  3. Fig 5 B-I: it is not clear what a small rectangle insert is. Also, the scale is illegible.
  4. Methods: 4.13 is a description of RNA extraction and RT-PCR but no gene expression data. This section should be either deleted or PCR data need to be included.
  5. Authors names are followed by a superscript “a, b, #” but only “a” is explained.

Author Response

Reviewer 3

In this study the authors investigated the contribution of RGD motif in the neuroprotective and anti-inflammatory properties of osteopontin (OPN) in MCAO. They tested several OPN peptides of various length containing RGD, SLAY and scrambled motifs both in vivo and in vitro. They found that RGD has a key role in neuroprotective effects of OPN. The study is generally well done and well written. There are only minor points to address:

  1. Panels B, D in Fig 1 and A, C in Fig 2 should be larger as it is hard to see the infarct volume.

Response: By following reviewer’s comment, we changed formats of Fig.1 and Fig.2 in the revised manuscript, providing a better picture quality.                            

  1. Results section 2.6: In the last sentence a N-NAM is mentioned. Is this an error?

Response: Thank you for pointing this out. We corrected it in the revised manuscript.

  1. Fig 5 B-I: it is not clear what a small rectangle insert is. Also, the scale is illegible.

Response: Thank you for raising this point. We replaced Figure 5B-I with a new set of images in the revised manuscript.

  1. Methods: 4.13 is a description of RNA extraction and RT-PCR but no gene expression data.5. This section should be either deleted or PCR data need to be included.

Response: Levels of M1 and M2 markers were examined using RT-PCR and the results were in Figure 5L-O.

  1. Authors names are followed by a superscript “a, b, #” but only “a” is explained.

Response: Thank you for raising this point. We corrected it in the revised manuscript.

Round 2

Reviewer 1 Report

Accept as it.

Reviewer 2 Report

I have no further questions